# Simultaneous Detection of Exosomal microRNAs Isolated from Cancer Cells Using Surface Acoustic Wave Sensor Array with High Sensitivity and Reproducibility

**DOI:** 10.3390/mi15020249

**Published:** 2024-02-07

**Authors:** Su Bin Han, Soo Suk Lee

**Affiliations:** Department of Pharmaceutical Engineering, Soonchunhyang University, 22 Soonchunhyang-ro, Shinchang-myeon, Asan-si 31538, Chungcheongnam-do, Republic of Korea; 1gkstkfkd@naver.com

**Keywords:** surface acoustic wave, sensor array, microRNA, titanium oxide nanoparticles, photocatalytic silver staining, signal amplification

## Abstract

We present a surface acoustic wave (SAW) sensor array for microRNA (miRNA) detection that utilizes photocatalytic silver staining on titanium dioxide (TiO_2_) nanoparticles as a signal enhancement technique for high sensitivity with an internal reference sensor for high reproducibility. A sandwich hybridization was performed on working sensors of the SAW sensor array that could simultaneously capture and detect three miRNAs (miRNA-21, miRNA-106b, and miRNA-155) known to be upregulated in cancer. Sensor responses due to signal amplification varied depending on the concentration of synthetic miRNAs. It was confirmed that normalization (a ratio of working sensor response to reference sensor response) screened out background interferences by manipulating data and minimized non-uniformity in the photocatalytic silver staining step by suppressing disturbances to both working sensor signal and reference sensor signal. Finally, we were able to successfully detect target miRNAs in cancer cell-derived exosomal miRNAs with performance comparable to the detection of synthetic miRNAs.

## 1. Introduction

Sensitive and highly reproducible detection technologies for biomolecules are essential for clinical and basic research applications [1,2,3]. Typical biosensing platforms use labels such as enzymes or fluorophores. However, the labeling process requires a long sample preparation time and additional cost [4,5,6]. To alleviate these concerns, label-free biosensing techniques such as surface plasmon resonance (SPR) [7,8], quartz crystal microbalance (QCM) [9,10,11,12,13], and surface acoustic wave (SAW) [14,15,16,17,18,19] have been developed. Among them, SAW sensors have been widely studied for detecting various target biomaterials due to their high sensitivity and reliability. In particular, the Love wave sensor (also known as guided shear horizontal SAW sensor) consisting of a piezoelectric substrate with a waveguide layer having low shear wave velocity, is one of the most promising sensing platforms with great potential for biosensor applications due to its high sensitivity and stability in a liquid phase [20,21,22]. In biosensors, the waveguide layer can protect the interdigital transducer (IDT) electrode from a liquid environment. The waveguide layer can also confine acoustic energy near the sensing surface, providing high sensing response to any physical perturbations on the surface, such as changes in mass density, mechanical stiffness, pressure, temperature, and viscosity. Various dielectric materials such as silicon dioxide (SiO_2_) [23,24], zinc oxide (ZnO) [25], and parylene polymer [26] can be used as waveguide materials. SiO_2_ is the most widely used waveguide layer due to its low acoustic loss, high mechanical and chemical resistance, and ease of functionalization with biomolecules.

SAW sensors adopt the principle that when mass is loaded on the sensor surface, a change in surface acoustic velocity occurs, which can be detected as a frequency shift or phase shift of the surface acoustic wave. Therefore, the frequency or phase shift of a SAW sensor is proportional to the accumulated mass on the sensor surface. In a previous study, we have demonstrated that piezoelectric biosensors combined with gold staining signal enhancement strategies exhibit high sensitivity through a tremendous increase in mass [16,17,18]. Although gold staining methods have been shown to be able to enhance signal intensities, non-uniform growth of gold nanoparticles may result in lower reproducibility. Additionally, to achieve better reproducibility, a method was needed to overcome background interference, such as non-specific adsorption of human serum proteins to sensor surface. As a result, it is necessary to introduce internal reference sensors into various biosensing platforms to increase reproducibility. This enables normalized data acquisition from working sensors and reference sensor signals, which can compensate any noises and distinguish non-specific binding events on the sensor surface. Additionally, normalization (working sensor response divided by reference sensor response) can be used to suppress perturbations known to affect both working and reference sensor signals similarly.

Exosomes are cell-derived small (30–90 nm) extracellular vesicles that promote intercellular communication and immunoregulatory functions. These vesicles shuttle various molecules, including miRNAs, to recipient cells. One of the most recent exciting findings is that microRNAs (miRNAs) exist in exosomes and these exosomal miRNAs can be functionally delivered to target cells. Thus, exosomal miRNAs are considered as biomarkers for many pathological states [27,28]. Exosomes from diseased individuals contain miRNAs not found in normal, healthy subjects [29,30]. MicroRNAs (miRNAs) are small (21–25 nucleotides in length), endogenous, non-coding, and naturally occurring single-stranded RNA molecules. They were first discovered in *Caenorhabditis elegans* in 1993 [31,32,33]. It has been reported that they can regulate the expression of other genes in various animals, plants, and viruses by binding to the 3′-untranslated region (3′-UTRs) of specific messenger RNAs. Several previous studies have reported that abnormal expression of specific miRNAs is involved in cancer, infectious diseases, vascular diseases, and immune disorders [34,35,36]. In particular, it is closely related to the occurrence, development, and spread of cancer. Therefore, miRNAs can be used as biomarkers for early detection of cancer and various diseases. They are also valuable therapeutic targets for treating diseases [37,38,39,40,41,42].

In this study, we describe highly sensitive 200 MHz Love wave SAW sensors capable of simultaneously detecting microRNA-21 (miR-21), microRNA-106b (miR-106b) and microRNA-155 (miR-155) whose up-regulation is known to be closely associated with cancer. Research has been conducted to analyze target DNA using SAW sensors [43,44], but no studies on miRNA detection have been reported yet. Therefore, research on detecting miRNA using SAW sensors is of great significance and becomes the starting point of this study. The Love wave SAW sensors consist of three working sensors and one adjacent reference sensor. A sandwich hybridization in combination with titanium dioxide-based photocatalytic silver staining was used as the basic detection method. Capture nucleotides and oligo T’s designed to capture three types of cancer-related miRNAs (miR-21, miR-106b and miR-155) were immobilized on working sensor regions and the reference sensor region of the SAW biosensor array, respectively. Each of the three miRNAs could hybridize with complementary capture nucleotides. Complete sandwich hybridization was then achieved by introducing a mixture of TiO_2_ nanoparticles conjugated with universal detection nucleotides and oligo A’s conjugated TiO_2_ nanoparticles. Subsequently, photocatalytic deposition of metallic silver on the surface of TiO_2_ nanoparticles was performed, leading to signal enhancement due to increased mass. This significantly increased the sensitivity of miRNA analysis by reducing the limit of detection (LOD) for three miRNAs as potential cancer biomarkers. Metal staining technique has been proven to enhance signals for detecting various analytes due to its superb sensitivity [45,46]. In particular, mass loading caused by metal staining can lead to dramatic signal enhancement in piezoelectric sensors. Although other signal amplification techniques, such as rolling circle amplification, can also enable highly sensitive detection of various target nucleotides, metal staining has several advantages. For example, metal staining does not require expensive, complicated, or multistep processes, making it adaptable to a low-cost and robust protocol. In this study, photocatalytic silver staining onto TiO_2_ nanoparticles was used as a robust signal enhancement technique to detect small amounts of target miRNAs. In addition, the reproducibility of analyzing a target analyte was significantly improved by introducing a reference sensor adjacent to working sensors in a SAW sensor array for the purpose of normalizing working sensor signal to reference sensor signal. The normalization of signals obtained from working sensors and a reference sensor built on a single substrate can minimize deviations that occur due to environmental factors (such as temperature, pressure, and viscosity) or non-specific adsorption and interference from serum proteins under the same conditions. Due to improved sensor reproducibility, the coefficients of variation (CVs) in sensor signals are dramatically reduced compared to those before normalization.

## 2. Materials and Methods

### 2.1. Reagents and Apparatus

HPLC purified synthetic miRNAs (miR-21, miR-106b, and miR-155) and 5′-amine modified oligonucleotide capture probes including complementary sequences of miR-21, miR-106b, and miR-155 were obtained from Bioneer (Daejeon, Korea) in lyophilized forms. A 3′-thiol modified universal detecting oligonucleotide probe, a 5′-amine modified oligo T’s reference capture probe, and a 3′-thiol modified oligo A’s reference detecting probe were also purchased from Bioneer. Their sequences are as listed in Table 1. Titanium oxide (particle size 21 nm, surface area 35–65 m^2^/g), silver nitrate, 3-glycidoxypropyl) triethoxysilane (3-GPTES), 6-amino-1-propanol, human serum, and saline sodium citrate (SSC) buffer (20X) were obtained from Sigma-Aldrich Chemical Co. (St. Louis, MO, USA). MCF-7 human breast cancer cell line (ATCC^®^ HTB-22™) was obtained from the American Type Culture Collection (ATCC, Rockville, MD, USA). Total Exosome Isolation reagent (TEI) was obtained from Invitrogen (Carlsbad, CA, USA). RNeasy Mini Kit for extracting total RNA was brought from Qiagen (Valencia, CA, USA). Ethanol, tetrahydrofuran (THF), *N*,*N*-dimethylformamide (DMF), and other organic solvents were purchased from Samchun Chemicals & Metals Co., Ltd. (Seoul, South Korea). All aqueous solutions were prepared in RNase-free water obtained from Thermo Fisher Scientific (Waltham, MA, USA).

### 2.2. Design and Fabrication of SAW Sensor Array

The SAW sensor array was designed as follows. First, two pairs of IDT electrodes were patterned onto a 36° YX-LiTaO_3_ piezoelectric substrate (Yamaju Ceramics Co. Ltd., Anada-Cho Seto City, Japan), a widely used piezoelectric material due to its large electromechanical coupling factor (*K*^2^) and low propagation and insertion loss [47]. Aluminum input and output IDT electrodes consisted of 72 finger electrode pairs with a width of 5.0 μm and a center-to-center separation of 10.0 μm. The spacing between delay lines was 2 mm (100 λ, λ = 20 μm). Aluminum thin film having a thickness of 3000 Å was sputtered onto the LiTaO_3_ wafers, patterned with a conventional photolithographic technique, and wet-etched to define the IDT electrodes. The area of the SAW sensor was 3.0 mm × 9.0 mm. The aperture of IDT electrodes was 1.6 mm. To confine the acoustic energy near the surface and protect the electrode from the buffer solution, a simulation-based 5.2 μm thick SiO_2_ guide layer was deposited on the sensing surface by plasma-enhanced chemical vapor deposition (P-500, Applied Materials, Inc., Santa Clara, CA, USA) on the wafer, and then a patterned chromium layer was applied on the SiO_2_ layer as an etch mask. To open contact pads for electrical connection, wet etching with buffered oxide etchant was then performed. The SAW sensor manufactured in this way could operate at a canter frequency of approximately 200 MHz. Four diced SAW sensors were mounted on a printed circuit board (PCB) and bonded with aluminum wires for electrical connection. Detailed configuration of the SAW sensor array is shown in Figure 1.

### 2.3. Sensing and Fluidic Blocks of the SAW Sensor Array

The SAW sensor array system consisted of a custom-made oscillator, a frequency counter, and two multiplexers with channel controller as shown in Figure 2. SAW sensors were positioned as a frequency determining element of the oscillator. Additional use of multiplexers enabled the sequential switching of the four SAW sensors by operating with the channel controller. Surface acoustic waves, which were passed through delay lines, were transmitted to the oscillator and fed back to the sensors again. The frequency counter measured the frequency of SAW signals using a field-programmable gate array (FPGA). The flow cell was constructed as shown in Figure 3 with a peristaltic pump (ISM597; ISMATEC, Glattbrugg, Switzerland), a custom-made polymethyl methacrylate (PMMA) fluidic block, and silicon gasket. In the array sensor, each spot was separated from other spots with a silicon gasket, which allowed for separate reaction chambers. Each spot was also connected to the peristaltic pump via a tube. To exclude effects of temperature changes in SAW sensor signals, a temperature controller was installed below the chamber to keep the temperature at 25 °C. The flow rate was kept at 1.0 mL/min and the volume of each reaction chamber was 30 μL. After each run, reaction chambers and silicone gaskets were thoroughly rinsed with 0.05% Tween 20 (Sigma-Aldrich, St. Louis, MO, USA) in SSC buffer solution (Invitrogen, Carlsbad, CA, USA) and double distilled water. The SAW sensor chip can be reused by thorough plasma cleaning but was not reused in this study.

### 2.4. Immobilization of Capture Probes on SiO_2_—Coated SAW Sensor Array and Conjugation of Detecting Probes on TiO_2_ Nanoparticles

Capture probe immobilization on the sensor surface: Silicon dioxide (SiO_2_)-coated SAW sensor arrays were sequentially cleaned with deionized water and ethanol. They were then dried under a nitrogen atmosphere and activated in a UV-ozone chamber (144AX-220; Jelight Company Inc., Irvine, CA, USA) for 5 min, followed by incubation in 3% (vol./vol.) 3-GPTES in ethanol for 1 h. After washing with ethanol and drying under nitrogen, they were baked at 110 °C in an oven for 1 h, washed again with ethanol, and dried under nitrogen. Next, 5′-amine-modified DNA capture probes were attached to the surface of 3-GPTES-modified SAW sensor arrays according to the following protocol: 3-GPTES-modified SAW sensor arrays were treated with 100 µM (concentration adjusted for high-density immobilization, Appendix A) of 5′-amine-modified three oligonucleotide capture probes and oligo T reference probe dissolved in 100 mM sodium phosphate buffer (pH 8.5) in separate spots for 60 min at 37 °C. The immobilization process of DNA capture probes on the SAW sensor chip was performed inside a humidity chamber to avoid evaporation. After washing with sodium phosphate-buffered solution, the unreacted epoxide groups of 3-GPTES on the sensing area were deactivated through treatment with 50 mM 6-amino-1-hexanol in 100 mM sodium phosphate buffer (pH 8.5) for 30 min at 37 °C. Freshly modified sensor arrays were washed with sodium phosphate buffer and finally with doubly distilled water. Capture probe-modified SAW sensor arrays were desiccated at room temperature for storage until use.

Detecting probes conjugation on the TiO_2_ nanoparticles: The 3-GPTES modification process and conjugation process with 3′-amine-modified universal detecting probe (100 µM) and 3′-amine-modified oligo A reference detecting probe (100 µM) were the same as above. Detecting probe-modified TiO_2_ nanoparticles were stored in nuclease-free water at 4 °C until use.

### 2.5. Sandwich Hybridization with Photocatalytic Silver Staining

To detect miRNAs (miR-21, miR-106b and miR-155) using the SAW sensor array, a mixed solution of spiked synthetic three miRNAs in human serum at various concentrations (0.1 pM to 1.0 μM) was introduced to each sensor surface of SAW sensor array and allowed to stand at room temperature for 5 min to enable the formation of a hybrid duplex between miRNAs and the capture oligonucleotide probe on the surface. A mixed solution of universal oligonucleotide detecting probe (0.1 pM to 1.0 μM) and oligo A’s reference detecting probe (100 pM) both conjugated with TiO_2_ nanoparticles in human serum was introduced to partially hybridized sensor surfaces and allowed to stand at 25 °C for 5 min for a complete sandwich hybridization. After washing with human serum for 1 min, a silver nitrate (10 mM) in SSC buffer (1X) solution was added to each sensor surface and irradiated with UV light using a UV hand lamp with a wavelength of 365 nm (Vilber Lourmat, Collégien, France) equipped with 4 W UV discharge tubes to induce silver staining reaction through photocatalytic reduction. After 2 min exposure, the SAW sensor array was then finally washed with 1X SSC buffer solution. All experiments were repeated in quadruplicates.

### 2.6. Detection of miRNAs Using Exosomal miRNAs Extracted from a MCF-7 Human Breast Carcinoma Cell Line

Exosomes were isolated from an MCF-7 human breast carcinoma cell line using the Total Exosome Isolation reagent (TEI) (Invitrogen, Carlsbad, CA, USA) and miRNAs were extracted using the RNeasy Mini Kit (Qiagen) following each manufacturer’s instructions. Exosomal miRNA samples obtained were diluted to various concentrations (1.0 pg/mL to 100 μg/mL). For the SAW sensor array measurements of exosomal miRNAs, the same procedure described above was performed. Various concentrations of exosomal miRNAs, ranging from 1.0 pg/mL to 100 μg/mL, were used instead of synthetic miRNAs. All experiments were repeated in triplicates.

## 3. Results

### 3.1. Detection of miRNAs by Sandwich Hybridization and Photocatalytic Silver Staining

Figure 4 highlights the sandwich hybridization process performed in this study, consisting of the immobilization of capture oligonucleotide probes, partial hybridization between capture oligonucleotide probes and target miRNAs in human serum, complete sandwich hybridization between target miRNAs and two complementary probes after introducing the universal detecting oligonucleotide probe with attached TiO_2_ nanoparticles and subsequent size enlargement of the TiO_2_ nanoparticles using photocatalytic silver staining strategy. Capture oligonucleotide probes were immobilized on the 3-GPTES coated SiO_2_ guiding layer. A major concern inherent in target biomolecule detection assays is a potential background interference from other biomolecules and chemical species. Provided that target miRNAs were present in the human sera, it was important to assess the impact of other human serum proteins in our detection strategy. Accordingly, all experiments were performed with human serum spiked with three miRNAs (miR-21, miR-106b and miR-155). The miRNAs present in human serum were hybridized with capture probes and subsequently combined with universal detection probes conjugated to TiO_2_ nanoparticles in a traditional sandwich hybridization format (Appendix A). The introduction of silver nitrate solution and UV irradiation resulted in photocatalytic deposition of silver onto captured TiO_2_ nanoparticles on the sensor surface.

Figure 5A shows sensor response to the sandwich hybridization between miR-21 (the concentration of miR-21 was 10 ng/mL) and capture probes and the universal detecting oligonucleotide probe with attached TiO_2_ nanoparticles followed by photocatalytic silver staining. As expected, the sandwich hybridization on the sensor surface caused a decrease in frequency due to a mass increase due to participating oligonucleotides and TiO_2_ nanoparticles. Additionally, despite coating the surface with 6-amino-1-hexanol, the use of human serum as a reaction medium resulted in a decrease in frequency due to non-specific adsorption of human serum proteins to the functionalized surface. Washing with SSC buffer showed a slight increase in frequency. This frequency change was indicative of incompletely absorbed human serum proteins on the functionalized surface, which were easily removed by washing with SSC solution. Most notably, the photocatalytic silver staining process resulted in an even greater decrease in resonance frequency. The resonance frequency decreased rapidly over time because photocatalytic deposition of metallic silver on TiO_2_ nanoparticles captured by sandwich hybridization resulted in a significant mass increase at the SAW sensor surface (Appendix A). As a result of photocatalytic silver staining, metallic silver nanocomposites were deposited irregularly on the TiO_2_ surface. Since the density of metallic silver (d = 10.51 g/cm^3^) was higher than that of TiO_2_ (d = 4.23 g/cm^3^), the mass loading effect was significant. From this perspective, we carried out atomic force microscopy (AFM), transmission electron microscopy (TEM), and X-ray diffraction (XRD) analysis to characterize the SAW sensor surface after photocatalytic silver staining on TiO_2_ nanoparticles. Figure 5B shows atomic force microscopy (AFM) images of sandwich hybridization results, including TiO_2_ nanoparticles on glass slides after photocatalytic silver staining processes. For clear images, we analyzed results of a sandwich hybridization performed using commercial glass slides under the same conditions as on SiO_2_-coated SAW sensor. The 4 μm × 4 μm surface was scanned. The brighter spots in Figure 5B are silver deposited TiO_2_ nanoparticles. Figure 5C shows a typical TEM image for TiO_2_/Ag composites, in which Ag nanoparticles were deposited irregularly as dark spots on the surface of TiO_2_. Figure 5D shows XRD patterns for silver deposition on the surface of TiO_2_ nanoparticles. XRD patterns of bare TiO_2_ nanoparticles exhibited diffraction peaks of anatase TiO_2_ (JCPDS card no. 21-1272). The average crystallite size estimated from the most intense (101) peak at 2*θ* = 25.2° in XRD patterns of anatase TiO_2_ was about 21 nm using the Scherrer equation. After silver deposition (TiO_2_/Ag), it was confirmed that additional diffraction peaks appeared at 2*θ* = 38.1°, 44.3°, and 64.5° due to the influence of Ag nanoparticles present on the TiO_2_ surface. The diffraction peak of Ag at 38.1° overlapped significantly with the anatase TiO_2_ peak at 37.8°. Figure 5E shows the dynamic light scattering (DLS particle size distribution of TiO_2_ and TiO_2_/Ag nanocomposites. These results demonstrated the potential of the SAW sensor to perform effective detection of miRNAs. However, it is necessary to investigate sensor responses to changes in concentrations of miRNAs.

### 3.2. Evaluation of the Response of SAW Sensors on the Concentration of miRNAs

We analyzed the effects of the miRNA’s concentration on sensor response due to sandwich hybridization and photocatalytic silver staining in SAW sensor array. The human serum was spiked with the three miRNAs, enabling simultaneous detection of miR-21, miR-106b, and miR-155. Measurements were carried out five times for each concentration of three synthetic miRNAs spiked in human serum. The results are displayed in Figure 6A. These miRNA samples were prepared by 10-fold serial dilution from a 1.0 μM stock solution of each of the synthetic miRNAs. They were analyzed based on the assay process mentioned in the above section. As concentrations of three miRNAs increased from 0.1 pM to 1.0 μM, the sensor response due to sandwich hybridization and photocatalytic silver staining also logarithmically increased, indicating that the amount of miRNA-oligonucleotides hybrid duplexes-TiO_2_ complexes formed on the SAW sensor surface was proportional to the applied miRNA concentration. The higher the concentration of miRNAs, the more sandwich hybridization occurred between capture probes, TiO_2_-attached detecting probes, and target miRNAs, which ultimately increased the surface concentration of TiO_2_ nanoparticles. This increase in surface concentration of TiO_2_ nanoparticles not only caused a mass loading effect in itself, but also caused a greater increase in frequency change due to photocatalytic deposition of silver on TiO_2_ nanoparticles. The limits of detection (LODs) of miR-21, miR-106b, and miR-155 were 0.048 pM, 0.084 pM, and 0.062 pM, respectively. The LOD was determined based on the standard deviation of the blank sample measurement as well as a low sample concentration (LOD = μ_B_ + 1.645σ_B_ + 1.645σ_S_, where μ_B_ and σ_B_ were mean and standard deviation of blank sample measurements and σ_S_ was the standard deviation of the population of the low sample measurements) according to the definition by Shrivastava and Gupta [48]. Here, blank refers to the result of all other processes proceeding without adding the target miRNA to the capture probe and the delta (Δ) frequency value of blank was 1.7 ± 0.15 KHz (Appendix A). The LOD for the detection of three miRNAs can be clearly distinguished from samples with the lowest concentration present in human serum, 0.1 pM, in terms of frequency shifts. In addition, it showed a good linearity over the entire concentration range from 0.1 pM to 1.0 μM of miRNAs expressed in log scale. These results show that it has a competitive LOD and linearity range compared to the recently reported miRNA sensors (Table 2) [49,50,51,52,53,54,55,56,57,58,59]. Meanwhile, according to the well-known Sauerbrey’s equation (1), the signal intensity (delta frequency) can be increased by using a high-frequency sensor under the same target mass and sensor surface area [60], which means improved sensitivity. Where *f*_0_ is resonance frequency, Δ*f* is the change in frequency (Hz), Δ*m* is the mass change (g), *A* is piezoelectrically active crystal area (m^2^), *ρ_s_* and *μ_s_* are the mass density (g·m^−3^) and shear modulus (g·m^−1^·s^−2^) of the sensor surface, silicon dioxide, respectively. In this study, the SAW sensor with a center frequency of 200 MHz consisted of 72 finger pairs with input and output IDT electrode widths of 5.0 μm. In theory, in order for the center frequency to be 400 MHz, 36 finger pairs with an electrode width of 2.5 μm need to be formed. This means that the electrode spacing and finger pairs can be reduced by half. A SAW sensor with a higher center frequency can be expected to have improved sensitivity, but there may be concerns about an increase in noise due to the greater influence on the external environment.
(1)$$∆f=-[2{f_{0}}^{2}/{A(\rho_{s}\mu_{s})}^{1/2}]∆m$$

### 3.3. Effect of Normalization on the Reproducibility for Detecting miRNAs in SAW Sensor Array

On the reference sensor of the SAW sensor array, a capture probe [5′-NH_2_-(CH_2_)_6_-TTT TTT TTT T-3′] is immobilized, and a detecting probe [3′-NH_2_-(CH_2_)_6_-AAA AAA AAA A-5′] conjugated with TiO_2_ nanoparticles (fixed at 100 pM) is hybridized to this and a subsequent photocatalytic silver enhancement reaction occurs. The hybridization of poly A and poly T of the same length occurs perfectly in the reference sensor, and since the reference sensor is configured in a sensor array adjacent to the working sensors, side reactions such as non-specific adsorption that occur in the working sensors occur at the same level in the reference sensor. Thus, we investigated the dependence of normalized sensor responses due to sandwich hybridization and photocatalytic silver staining on applied concentrations of three miRNAs. The normalized results of five measurements for each concentration of miR-21, miR-106b, and miR-155 are shown in Figure 6B. We compared coefficients of variation (CVs) in normalized sensor responses with those of working sensor responses. CVs of normalized sensor responses were in the range of 1.2% to 3.4%, which was lower than those of working sensor responses in the range of 8.1% to 16.3%. In this assay, high standard deviations or CVs of working sensor responses mainly resulted from background noise and non-linear growth of silver nanoparticles on TiO_2_ due to photocatalytic silver staining. Therefore, introduction of the internal reference sensor reduced CVs of assays, improving the reproducibility of the SAW sensor array. This demonstrated that normalization could screen out background noise by manipulating data and minimize non-uniformity in the photocatalytic silver staining process by suppressing disturbances to both working sensor signals and reference sensor signal. A linear log-logit transformation was used to fit the calibration curve. Normalized signals were converted into concentrations using calibration curves. The LODs of miR-21, miR-106b, and miR-155 using normalized signals were 0.012 pM, 0.026 pM, and 0.015 pM, respectively. This indicates that the introduction of the internal reference sensor with the normalization process can also improve the LODs of the SAW sensor array for the detection of miRNAs. Hence, it was confirmed that the SAW sensor array could carry out selective, sensitive and reproducible detection of miRNAs. The good target selectivity of the SAW sensor for miRNAs is depicted in the Appendix A).

### 3.4. Detection of miR-21, miR-106b and miR-155 in Cancer Cell-Derived Exosomal miRNA Samples Using SAW Sensor Array

To evaluate the performance of the SAW sensor array for analyzing exosomal miRNA samples derived from cancer cells, we performed detection assays of miR-21, miR-106b, and miR-155 using exosomal miRNAs extracted from exosomes isolated from MCF-7 cells, a breast cancer research model [61]. The MCF-7 cell line is routinely used as a model system for studying human breast cancer. It is known that upregulation of miR-21, miR-106b, and miR-155 promotes the proliferative ability of MCF-7 cells in vivo [62]. Exosomes were isolated from MCF-7 cells using the Total Exosome Isolation reagent (Invitrogen) and miRNAs were extracted using the RNeasy Mini Kit (Qiagen), which were serially diluted 10-fold to obtain various concentrations (1.0 pg/mL to 100 μg/mL). Exosomal miRNAs samples thus prepared were then applied to the SAW sensor array in the same manner as synthetic miRNAs described in the previous section. Experiments were repeated four times for each exosomal miRNA sample in the concentration range of 1.0 pg/mL to 100 μg/mL. The results are shown in Figure 7A. The SAW sensor array showed a good linearity when expressing miRNA concentration in a logarithmic scale. The LODs in the present linear ranges were 0.67 pg/mL, 0.79 pg/mL, and 0.84 pg/mL for miR-21, miR-106b, and miR-155, respectively. However, the LODs for miR-21, miR-106b, and miR-155 using normalized signals were 0.21 pg/mL, 0.37 pg/mL, and 0.42 pg/mL, respectively (Figure 7B). In addition, CVs of normalized sensor responses were in the range of 1.8% to 3.8%, which were much lower than those of working sensor responses in the range of 10.4% to 24.8%. This once again proved that the improvement in the LODs for target miRNAs in cancer cell-derived miRNAs through the normalization process was due to the introduction of an internal reference sensor. As a result of detecting the three target miRNAs in exosomal miRNAs extracted from MCF-7 cancer cells, designed miRNA detection strategies of the SAW sensor array worked sufficiently for cancer cell-derived exosomal miRNA samples.

## 4. Conclusions

In conclusion, we demonstrated a Love wave SAW sensor array including an internal reference sensor in combination with signal enhancement strategy using photocatalytic silver staining on TiO_2_ nanoparticles for simultaneous detection of miRNAs in a sensitive and reproducible manner. A sandwich hybridization format was utilized. After miRNAs were partially hybridized with capture probes immobilized on the sensor surface, complete sandwich hybridization then occurred upon addition universal detecting probe conjugated with TiO_2_ nanoparticles. Subsequent photocatalytic silver staining resulted in signal enhancement. It was confirmed that normalization (a ratio of working sensor response to reference sensor response) could screen out background interferences by manipulating data and minimize non-uniformity in the photocatalytic silver staining step by suppressing disturbances to both the working sensor signal and reference sensor signal. We also showed that normalized sensor response depended on both the concentration of the synthetic miRNAs and the concentration of exosomal miRNAs derived from MCF-7 cancer cells. Our future efforts will be focused on extending this platform to detection of other disease markers such as proteins and small molecules present in body fluids.

Due to its size, sensitivity, and reliability, this SAW sensor array platform is expected to be useful for developing sensing devices for point-of-care diagnostics.

## Figures and Tables

**Figure 1 micromachines-15-00249-f001:**
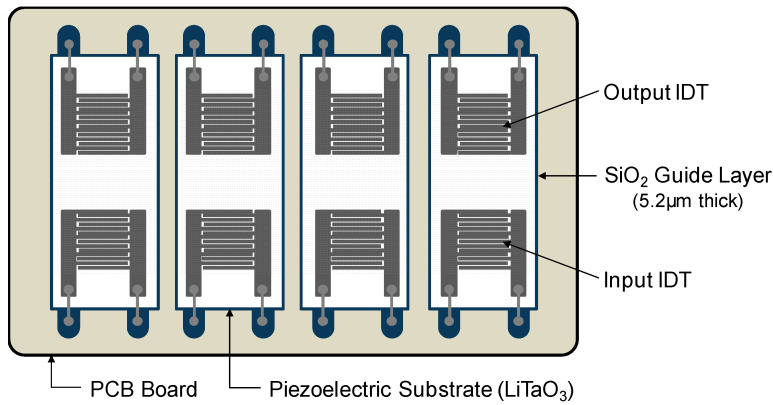
Top view of the Love wave SAW sensor array. Four SAW sensors were mounted on PCB and bonded with aluminum wires for electrical connection.

**Figure 2 micromachines-15-00249-f002:**
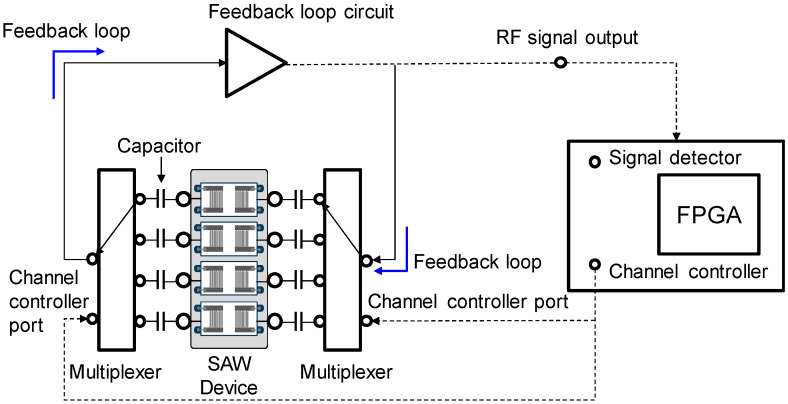
Block diagram of the measuring system for the SAW sensor array consisting of a custom-made oscillator, a frequency counter, and two multiplexers with channel controller.

**Figure 3 micromachines-15-00249-f003:**
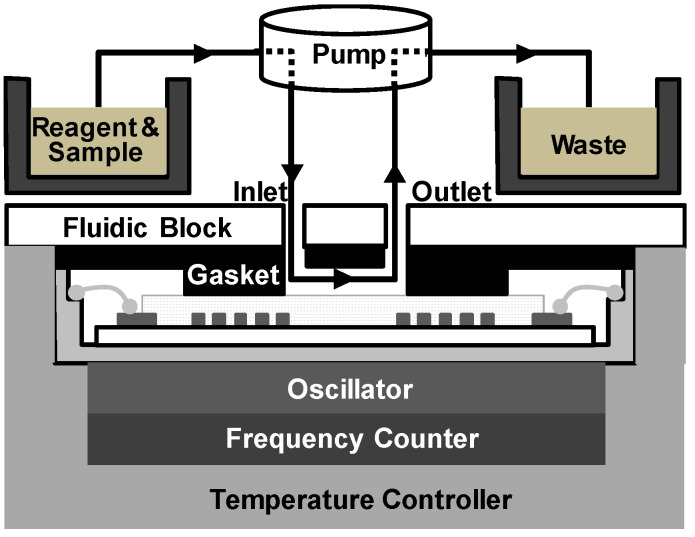
Configuration of the fluidic cell consisting of a peristaltic pump for fluid transfer with a custom-made fluidic block and silicon gasket for fluid control.

**Figure 4 micromachines-15-00249-f004:**
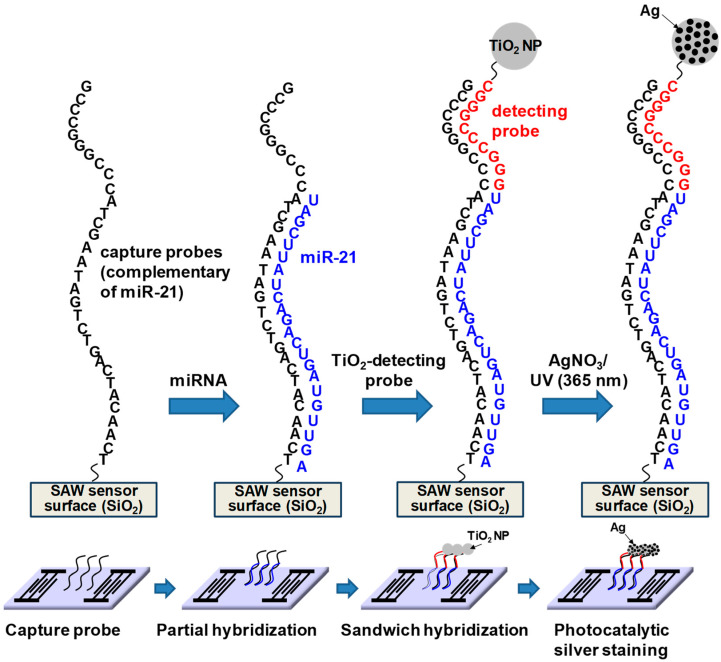
Schematic of the sandwich hybridization format utilized in this study in combination with TiO_2_-mediated photocatalytic silver staining on a SAW sensor.

**Figure 5 micromachines-15-00249-f005:**
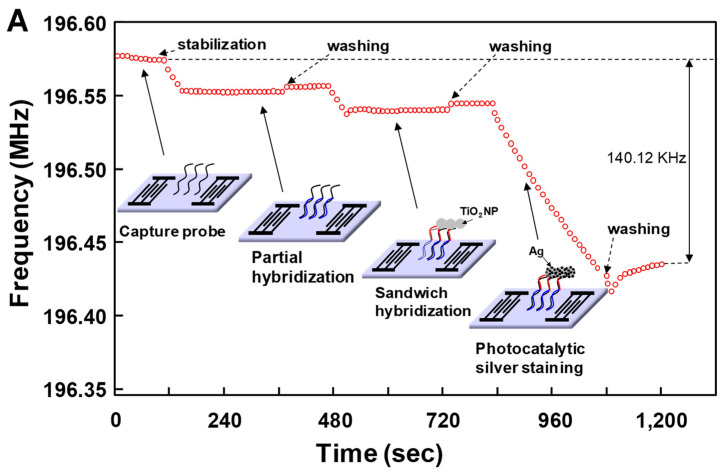
(**A**) Sensor response to the sandwich hybridization formation between the 10 nM of miR-21, a capture probe and a universal detecting oligonucleotide probe attached with TiO_2_ nanoparticle followed by photocatalytic silver staining, (**B**) AFM image of the sandwich hybridized complex of capture probe-miR-155-detecting probe conjugated with TiO_2_ nanoparticles after photocatalytic silver staining, (**C**) TEM images of the TiO_2_/Ag nanocomposites, (**D**) XRD patterns of anatase TiO_2_ and TiO_2_/Ag nanocomposites and (**E**) DLS particle size distribution of TiO_2_ and TiO_2_/Ag nanocomposites.

**Figure 6 micromachines-15-00249-f006:**
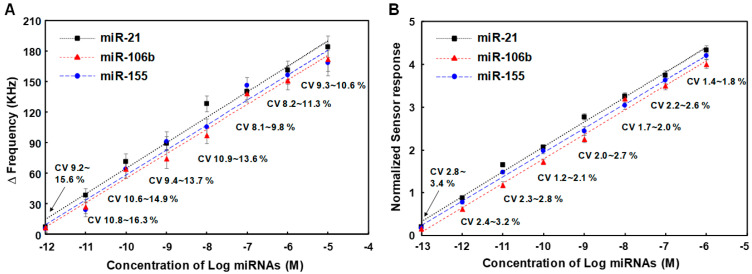
(**A**) Changes in resonance frequency of SAW sensor array according to concentrations of three synthetic miRNAs in the range of 0.1 pM to 1.0 μM, (**B**) Variation in normalized sensor responses due to changes in concentrations of three synthetic miRNAs spiked in human serum. Measurements were performed five times for each concentration of miRNAs and blank subtraction was performed on all delta (Δ) frequency values for each concentration of miRNAs.

**Figure 7 micromachines-15-00249-f007:**
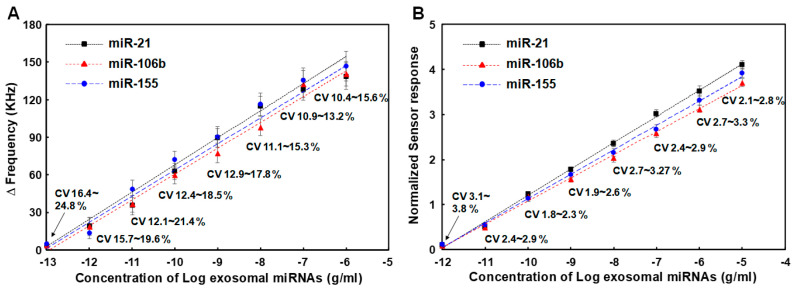
(**A**) Changes in resonance frequency of SAW sensor array according to concentrations of exosomal miRNAs from MCF-7 cancer cells in the range of 1.0 pg/mL to 100 μg/mL, (**B**) Variations in normalized sensor responses due to changed concentrations of three synthetic miRNAs spiked in human serum. Measurements were performed five times for each concentration of miRNAs.

**Table 1 micromachines-15-00249-t001:** Oligonucleotide sequence used in this experiment.

Oligonucleotides	Sequences
miR-21	5′-UAG CUU AUC AGA CUG AUG UUG A-3′
miR-106b	5′-UAA AGU GCU GAC AGU GCA GAU-3′
miR-155	5′-UUA AUG CUA AUC GUG AUA GGG GUU-3′
capture probes (complementary sequence of miR-21)	5′-H_2_N-(CH_2_)_6_-TCA ACA TCA GTC TGA TAA GCT ACC CGG GCC CG-3′
capture probes (complementary sequence of miR-106b)	5′-H_2_N-(CH_2_)_6_-ATC TGC ACT GTC AGC ACT TTA CCC GGG CCC G-3′
capture probes (complementary sequence of miR-155)	5′-H_2_N-(CH_2_)_6_-AAC CCC TAT CAC GAT TAG CAT TAA CCC GGG CCC G-3′
reference probe	5′-H_2_N-(CH_2_)_6_-TTT TTT TTT T
universal detecting probe	5′-CGG GCC CGG G-(CH_2_)_6_-NH_2_-3′
reference detecting probe	5′-AAA AAA AAA A-(CH_2_)_6_-NH_2_-3′

**Table 2 micromachines-15-00249-t002:** Comparison of LOD for miRNA detection using various detection methods reported in the literature.

Analytical Techniques	Target miRNA	Linear Range (pM)	LOD (pM)	Ref.
Fluoresence	miR-21	1000–50,000	330	[43]
Fluoresence	miR-21	0–300,000	4500	[44]
Fluoresence	miR-106b	0.001–1000	0.00044	[45]
Electrochemical	miR-21	0.096–25	0.029	[46]
Electrochemical	miR-107	0–1000	0.0001	[47]
Electrochemical	miR-107	0.005–5	0.01	[48]
Electrochemical	miR-155	0.5–25,000	0.96	[49]
Electrochemical	miR-21	0.1–10	0.64	[50]
QCM	miR-21	1000–10,000	400	[51]
SPR	miR-21	0–1.8	0.047	[52]
SERS	miR-21	4440–1,480,000	850	[53]
SAW	miR-21	0.1–1,000,000	0.012	This work
SAW	miR-106b	0.1–1,000,000	0.026	“
SAW	miR-155	0.1–1,000,000	0.015	“

## Data Availability

Data are contained within the article.

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
