# Peer review of "Simultaneous Detection of Exosomal microRNAs Isolated from Cancer Cells Using Surface Acoustic Wave Sensor Array with High Sensitivity and Reproducibility"

_micromachines, 2024, doi:10.3390/mi15020249_

Round 1

Reviewer 1 Report

Comments and Suggestions for Authors

In this work, the authors proposed a surface acoustic wave (SAW) sensor array for the detection of microRNAs (miRNA) that uses photocatalytic silver staining on titanium dioxide (TiO2) nanoparticles as a detection technique. Signal enhancement for high sensitivity with an internal reference sensor for high reproducibility.

Before considering this manuscript for publication, the following points should be addressed:

° The choice of the sensors investigated should clearly be explained.

° The manuscript should be checked for grammatical errors and few typos

° the authors should give more details on the manufacture of SAW Example:  what manufacturing process is used to produce aluminum IDTs?  

° line 139-140:  Please indicate finger periodicity IDT's (lambda)

° A real photo of the Sensor SAW will be appreciated 

° other references may strengthen this work such as: the works on SAW Discriminating DNA

Author Response

In this work, the authors proposed a surface acoustic wave (SAW) sensor array for the detection of microRNAs (miRNA) that uses photocatalytic silver staining on titanium dioxide (TiO2) nanoparticles as a detection technique. Signal enhancement for high sensitivity with an internal reference sensor for high reproducibility.

Before considering this manuscript for publication, the following points should be addressed:

° The choice of the sensors investigated should clearly be explained.

Answer: Thank you very much for your suggestion. The SAW sensor is a representative piezoelectric sensor and has been used not only as a pressure and viscosity sensor but also for the detection of various chemical and biomolecules. As a biosensor, SAW device has shown promise in detecting DNA and proteins for the purpose of diagnostics of several disease. In particular, researches have been conducted to analyze target DNA using SAW sensors, but no studies on miRNA detection have been reported yet. Therefore, the study of miRNA detection using the SAW sensor was of great significance and became the starting point of this study. We have inserted the following into the introduction part: “Researches have been conducted to analyze target DNA using SAW sensors [43, 44], but no studies on miRNA detection have been reported yet. Therefore, the study of miRNA detection using the SAW sensor was of great significance and became the starting point of this study.”

° The manuscript should be checked for grammatical errors and few typos

Answer: Thank you very much for your point out. The manuscript has been proofread by a native speaker of English (Certificate of editing attached).

° The authors should give more details on the manufacture of SAW Example:  what manufacturing process is used to produce aluminum IDTs?  

Answer: Thank you very much for your suggestion. Section 2.2 provides detailed information on the SAW sensor fabrication, and in response to reviewer requests, some more detail has been added, including the formation of aluminum IDTs. Aluminium input and output IDT electrodes consisted of 72 finger electrode pairs with a width of 5.0 μm and a center-to-center separation of 10.0 μm. The spacing between delay lines was 2 mm (100 λ,  λ= 20 μm). Aluminum thin film having a thickness of 3000 Å was sputtered onto the LiTaO3 wafers, patterned with a conventional photolithographic technique, and wet-etched to define the IDT electrodes. The area of the SAW sensor was 3.0 mm × 9.0 mm. The aperture of IDT electrodes was 1.6 mm. To confine the acoustic energy near the surface and protect the electrode from the buffer solution, a simulation-based 5.2 μm thick SiO2 guide layer was deposited on the sensing surface by plasma-enhanced chemical vapor deposition (P-500, Applied Materials, Inc., Santa Clara, CA, USA) on the wafer, and then a patterned chromium layer was applied on the SiO2 layer as an etch mask. To open contact pads for electrical connection, wet etching with buffered oxide etchant was then performed. The SAW sensor manufactured in this way could operate at a canter frequency of approximately 200 MHz. Diced four SAW sensors were mounted on printed circuit board (PCB) and bonded with aluminum wires for electrical connection. Detailed configuration of the SAW sensor array is shown in Figure 1.

° line 139-140:  Please indicate finger periodicity IDT's (lambda)

Answer: Thank you very much for your point out. The finger periodicity of the IDTs determines the SAW wavelength, which was set to λ200 MHz SAW = 20 μm. This sentence has been corrected as follows: “The spacing between delay lines was 2 mm (100  λ,  λ= 20 μm).”

° A real photo of the Sensor SAW will be appreciated 

Answer: First of all, we apologize for not being able to provide real photos of the SAW sensor array. We custom-produced the SAW sensor array. A patent application has been completed jointly with the manufacturer for the single sensor and the array sensor of the previous version, but the latest version of the array sensor (the array sensor in this paper) has not been applied for, so the manufacturer is still opposed to providing it in the paper. Therefore, only single sensor and previous version array sensor photos can be provided. However, I will not include them in the paper because they are meaningless. (previous version array sensor photos attached)

° other references may strengthen this work such as: the works on SAW Discriminating DNA

Answer: Thank you very much for your suggestion. In response to your question 1, we have added information and references regarding previous DNA analysis using the SAW sensor. We have inserted the following into the introduction part: “Researches have been conducted to analyze target DNA using SAW sensors [43, 44], but no studies on miRNA detection have been reported yet. Therefore, the study of miRNA detection using the SAW sensor was of great significance and became the starting point of this study.”

  1. Huang, Y.; Das, P.Kr.; Bhethanabotla, V.R. Surface acoustic waves in biosensing applications, Sens. Actuators Rep. 2021, 3, 100041.
  2. Najla, F.; Mathieu, L.; Christophe, V.; Jean-Marie, F.; Chouki, Z.; Lionel, R.; Patrick, L.; Jean Jacques B.; and Christine, P. Surface Acoustic Waves Sensor for DNA-Biosensor Development, Sens. Lett. 2009, 7, 847-850.

Reviewer 2 Report

Comments and Suggestions for Authors

This paper introduces a surface acoustic wave (SAW) sensor array for simultaneously capturing and detecting three miRNAs (miRNA-21, miRNA-106b, and miRNA-155) known to be upregulated in cancer. The technology utilizes photocatalyzed silver staining of titanium dioxide (TiO2) nanoparticles as a highly sensitive signal enhancement technique to sandwich the working sensor of the SAW sensor array, which has high repeatability. Finally, target miRNAs in cancer-derived exosomal miRNAs can be successfully detected, and their performance is comparable to that of synthetic miRNAs. However, the detailed description of some original explanations and experiments is not yet perfect, so the following modifications need to be made before publication:

1.       DLS distribution curves of the prepared Ag and TiO2 nanoparticles should be provided to show the nanoparticle size.

2.       In Fig. 5D, the XRD characteristic peak of Ag+TiO2 appears at approximately 38. Could you please explain why? The XRD characteristic peak position of silver should be 44.765, 65.166, 82.531.

3.       Photocatalytic silver staining is mentioned in the article. What is the role played in this article? What is the mechanism?

4.       Non-specific adsorption will also affect the frequency response. How to avoid non-specific adsorption?

5.       Please provide selectivity data for this saw device for markers detection. The paper discussed the use of titanium dioxide (TiO2) nanoparticle photocatalyzed silver staining as a high-sensitivity signal enhancement technology but did not compare TiO2, Ag, and Ag+TiO2 nanoparticles. Can you provide more details about this as well?

6.       The symbol “A” is lost in Figure 5A.

Comments on the Quality of English Language

Minor editing of English language is required.

Author Response

This paper introduces a surface acoustic wave (SAW) sensor array for simultaneously capturing and detecting three miRNAs (miRNA-21, miRNA-106b, and miRNA-155) known to be upregulated in cancer. The technology utilizes photocatalyzed silver staining of titanium dioxide (TiO2) nanoparticles as a highly sensitive signal enhancement technique to sandwich the working sensor of the SAW sensor array, which has high repeatability. Finally, target miRNAs in cancer-derived exosomal miRNAs can be successfully detected, and their performance is comparable to that of synthetic miRNAs. However, the detailed description of some original explanations and experiments is not yet perfect, so the following modifications need to be made before publication:

  1.  DLS distribution curves of the prepared Ag and TiOnanoparticles should be provided to show the nanoparticle size.

Answer: Thank you very much for your suggestion. We have inserted the DLS particle size distribution of TiO2 and TiO2/Ag nanocomposites in Figure 5(E). (Figure 5(E) attached)

  1. In Fig. 5D, the XRD characteristic peak of Ag+TiO2appears at approximately 38. Could you please explain why? The XRD characteristic peak position of silver should be 44.765, 65.166, 82.531.

Answer: Thank you very much for your comment. In this study, as a result of silver deposition on the TiO2 surface, an improvement in peak intensity was observed. Due to the presence of Ag on the TiO2 surface, additional peaks appear about 2θ = 38.1° (111), 44.3° (200), and 64.5° (220). The main diffraction peak of Ag at 38.1° (111) cannot be observed independently due to significant overlap with the anatase TiO2 peak at 37.8°. The XRD peaks attributed to silver nanoparticles are in very good agreement with the results of several other previous research papers (please see the figures below). (Figures attached)

  1. Photocatalytic silver staining is mentioned in the article. What is the role played in this article? What is the mechanism?

Answer: Thank you very much for your comment. As shown in Figure 5(A) in the manuscript, the photocatalytic silver staining process resulted in the greatest decrease in resonance frequency among the entire processes. The resonance frequency decreased rapidly over time because photocatalytic deposition of metallic silver on TiO2 nanoparticles captured by sandwich hybridization resulted in a significant mass increase at the SAW sensor surface. This result occurs because the density of metallic silver (d = 10.51 g/cm3) is higher than that of TiO2 (d = 4.23 g/cm3), resulting in a large mass loading effect. That is, the silver staining on TiO2 nanoparticle caused a large change in resonance frequency due to the mass loading effect, ultimately leading to an improvement in sensitivity. These contents are already included in Section 3.1.

The photocatalytic silver staining reaction mechanism is as follows: When TiO2 semiconductor is illuminated with greater energy than a band gap, an electron will move from the valence band to the conduction band to produce holes  (h+vb) in the valence band and the electrons (e-cb) in the conduction band. The electrons will interact with the surrounding silver ion (Ag+ from AgNO3 in this experiments) and produce metallic silver (Ag0). This is TiO2-mediated photocatalytic reduction of silver ion (Ag+) to metallic silver (Ag0). (Figure attached)

  1. Non-specific adsorption will also affect the frequency response. How to avoid non-specific adsorption?

Answer: Thank you very much for your comment. As you said, non-specific adsorption affects the frequency change in piezoelectric sensors such as SAW sensors. To prevent non-specific adsorption, the sensor surface was coated with 6-amino-1-hexanol, which is commonly used in DNA sensors, after immobilization of the capture probe. Nevertheless, when the use of human serum as a reaction medium resulted in a decrease in frequency due to non-specific adsorption of human serum proteins to the functionalized surface. Washing with SSC buffer appears to slightly increase the frequency, but is not a main solution. In addition, non-specific adsorption may reduce reproducibility because the amount adsorbed is different for each experiment. Therefore, we performed blank subtraction (subtract the blank response from the response at a certain concentration) in all concentrations of miRNA experiments. Additionally, an internal reference sensor was introduced to increase reproducibility. This allows normalized data acquisition from working sensors and reference sensor signals (working sensor response divided by reference sensor response), which can compensate any noises and distinguish non-specific binding events on the sensor surface.

We have inserted the following sentence about blank subtraction into the caption of Figure 5: " blank subtraction was performed on all delta (D) frequency values ​​for each concentration of miRNAs". The details regarding normalization by internal reference sensors have already been written in Section 3.3.

  1. Please provide selectivity data for this saw device for markers detection. The paper discussed the use of titanium dioxide (TiO2) nanoparticle photocatalyzed silver staining as a high-sensitivity signal enhancement technology but did not compare TiO2, Ag, and Ag+TiO2nanoparticles. Can you provide more details about this as well?

Answer: Thank you very much for your suggestion. Hybridization of oligonucleotides is a sequence-specific reaction, so it occurs for probes with complementary sequences to each miRNA. Therefore, the selectivity for target miRNA in this study is very high. As an example, for 100 nM concentrations of miR-21, miR-106b, and miR-155, the selectivity for probe with sequences complementary to that of miRNA-21 was tested using a single SAW sensor, and the results are presented in the Supplementary Information (Figure S4). Thus, we described the selectivity of the SAW sensor array at the end of Section 3.3 as follows: Hence, it was confirmed that the SAW sensor array could carry out selective, sensitive and reproducible detection of miRNAs. The good target selectivity of the SAW sensor for miRNAs is depicted in the Supplementary Information (Figure S4). (Figure S4 attached)

Meanwhile, for the detection of miR-21 in a single SAW sensor, we tested using the following four detection probes as you suggested: without labeling, labeling of TiO2 nanoparticles (21 nm), labeling of silver nanoparticles (10 nm obtained from Sigma-Aldrich Chemical Co) and labeling of TiO2 nanoparticle and photocatalytic silver staining. The test results using 100 nM of miR-21 were summarized and are also included in the Supplementary Information (Figure S3). TiO2 nanoparticle labeling and photocatalytic silver staining resulted in the largest frequency change, which was greater than the frequency change caused by TiO2 and Ag labeling combined. This can be seen as the effect of a small but large number of Ag nanoparticles being deposited on the TiO2 surface, as shown in the TEM image in Figure 4(C) in the manuscript. (Figure S3 attached)

  1. The symbol “A” is lost in Figure 5A.

Answer: Thank you very much for your point out. We have inserted it.

Reviewer 3 Report

Comments and Suggestions for Authors

This work report highly sensitive 200 MHz Love wave SAW sensors capable of simultaneously detecting microRNA-21 (miR-21), microRNA-106b (miR-106b) and microRNA-155 (miR-155). These Love wave SAW sensors consist of three working sensors and one adjacent reference sensor. A sandwich hybridization in combination with titanium dioxide-based photocatalytic silver staining was used as the basic detection method. This work is useful for the community. However, I have several concerns before this manuscript can be accepted. Therefore, in its current form, minor revisions are needed.

1. The authors need to improve the image quality. Some images contain text that is too small, such as Fig 5 and 6.

2.Could the author make some comments regarding the structure design? For example, if you change the electrode sizes, what will this adjustment affects the device performance?

3. Could this chip be reused again? 

Author Response

This work report highly sensitive 200 MHz Love wave SAW sensors capable of simultaneously detecting microRNA-21 (miR-21), microRNA-106b (miR-106b) and microRNA-155 (miR-155). These Love wave SAW sensors consist of three working sensors and one adjacent reference sensor. A sandwich hybridization in combination with titanium dioxide-based photocatalytic silver staining was used as the basic detection method. This work is useful for the community. However, I have several concerns before this manuscript can be accepted. Therefore, in its current form, minor revisions are needed.

  1. The authors need to improve the image quality. Some images contain text that is too small, such as Fig 5 and 6. 

Answer: Thank you very much for your comment. We have adjusted the text size in Figures 5, 6, and 7 as you pointed out.

2. Could the author make some comments regarding the structure design? For example, if you change the electrode sizes, what will this adjustment affects the device performance?

Answer: Thank you very much for your comment. In this study, the input and output IDT electrodes consisted of 72 finger pairs with an electrode width of 5.0 μm to obtain a center frequency of 200 MHz. In order for the center frequency to be 400 MHz, it must be composed of 36 finger pairs with an electrode width of 2.5 μm. In other words, the electrode spacing and finger pairs can be reduced by half. The signal intensity (delta frequency) can be increased by using a high-frequency sensor under the same target mass and sensor surface area by the well-known Sauerbrey's equation (1). (Equation attached)

That is, by reducing the thickness and spacing of IDT electrodes in our SAW sensor, high frequencies can be obtained and thus larger changes in frequency (delta frequency) can be obtained under the same mass and sensor surface area. Although the SAW sensor with a higher center frequency can be expected to have improved sensitivity, there may be concerns about an increase in noise due to the greater influence on the external environment. We have inserted the following into the end of Section 3.2: Meanwhile, according to the well-known Sauerbrey’s equation (1), the signal intensity (delta frequency) can be increased by using a high-frequency sensor under the same target mass and sensor surface area [60], which means improved sensitivity. Where f0 is resonance frequency, Df is the change in frequency (Hz), Dm is the mass change (g), A is piezoelectrically active crystal area (m2), rs and ms are the mass density (g·m-3) and shear modulus (g·m-1·s-2) of the sensor surface, silicon dioxide, respectively. In this study, SAW sensor with a center frequency of 200 MHz consisted of 72 finger pairs with input and output IDT electrode widths of 5.0 μm. In theory, in order for the center frequency to be 400 MHz, 36 finger pairs with an electrode width of 2.5μm need to be formed. This means that the electrode spacing and finger pairs can be reduced by half. A SAW sensor with a higher center frequency can be expected to have improved sensitivity, but there may be concerns about an increase in noise due to the greater influence on the external environment. (Equation attached)

  1. Could this chip be reused again?

Answer: Our SAW sensors can be reused by thoroughly cleaning them after use. Cleaning methods include plasma exposure and piranha treatment, but the latter is not preferred due to the risk of the substance itself and the risk of damage to aluminum wires for electrical connections. We have inserted the following into the end of Section 2.3: The SAW sensor chip can be reused by thorough the plasma cleaning, but was not reused in this study.

Reviewer 4 Report

Comments and Suggestions for Authors

Authors reported a surface acoustic wave (SAW) sensor array for microRNA (miRNA) detection with high sensitivity and reproducibility. Authors utilized photocatalytic silver staining on titanium dioxide (TiO2) nanoparticles as a signal enhancement technique for high sensitivity, and an internal reference sensor was applied for the high reproducibility. A sandwich hybridization was performed on the SAW sensor array which achieves three miRNAs (miRNA-21, miRNA-106b, and miRNA-155) simultaneously detection. This sensor array can be applied to detect target miRNAs in cancer cell-derived exosomal miRNAs. The experimental design is reasonable and interesting. Therefore, I recommend this paper to be major revised before acceptance.

Comments and questions:

1.     In the introduction, the author only emphasized the importance of internal reference sensor but did not introduce how the internal reference sensor work in this paper.

2.     The title is “Simultaneous Detection of Exosomal microRNAs Isolated from Cancer Cells Using Surface Acoustic Wave Sensor Array with High Sensitivity and Reproducibility”, but there is no information about exosomal microRNAs in the introduction, why did you choose exosomal microRNAs as target? Why the exosomal microRNAs detection is important?

3.     What’s the “4” in line 65 before “Several previous studies…”? The other small suggestion is that there is no need to introduce too much experiment detail in the introduction, for example in line 80 “ …serum was introduced into the sensor array and incubated for 10 minutes…”.

4.     It will be clearer if list the sequences information in a table.

5.     Supplement the details of probes immobilization and hybridization. Better to draw the chemical reaction step by step, and how probes hybridize with each other with probes sequences.

6.     For Figure 4, change the probes and target RNA with different color.

7.     The sandwich hybridization should be validated at least by gel assay. The length and concentration of probes should be optimized.

8.     For Figure 5, there is no A label in the left top of the figure, the result of the sensor without target RNA should be tested.

9.     For table 1, this work should also be compared with other SAW based miRNA detection methods.

In summary, I recommend this paper to be majorly revised before accepting.

Comments on the Quality of English Language

Logic needs to be improved. 

Author Response

Authors reported a surface acoustic wave (SAW) sensor array for microRNA (miRNA) detection with high sensitivity and reproducibility. Authors utilized photocatalytic silver staining on titanium dioxide (TiO2) nanoparticles as a signal enhancement technique for high sensitivity, and an internal reference sensor was applied for the high reproducibility. A sandwich hybridization was performed on the SAW sensor array which achieves three miRNAs (miRNA-21, miRNA-106b, and miRNA-155) simultaneously detection. This sensor array can be applied to detect target miRNAs in cancer cell-derived exosomal miRNAs. The experimental design is reasonable and interesting. Therefore, I recommend this paper to be major revised before acceptance.

Comments and questions:

  1. In the introduction, the author only emphasized the importance of internal reference sensor but did not introduce how the internal reference sensor work in this paper. (normalization)

Answer: Thank you very much for your comment. In Section 3.3 and 3.4, we described the improved reproducibility (% CV) resulting from normalization. In this study, normalization refers to the working sensor response (resonance frequency change) divided by reference sensor response. Since three working sensors and a reference sensor are setting in one sensor array, they operate under the same experimental conditions and can be said to be at the same level with respect to errors resulting from environmental factors or non-specific adsorption on the sensor surface in repeated experiments. Therefore, as described in the introduction part, the normalized data acquisition from working sensors and reference sensor signals allows compensating for any noise and distinguishing non-specific binding events on the sensor surface. Normalization can also be used to suppress perturbations known to affect both working and reference sensor signals similarly.

  1. The title is “Simultaneous Detection of Exosomal microRNAs Isolated from Cancer Cells Using Surface Acoustic Wave Sensor Array with High Sensitivity and Reproducibility”, but there is no information about exosomal microRNAs in the introduction, why did you choose exosomal microRNAs as target? Why the exosomal microRNAs detection is important?

Answer: Thank you very much for your comment. Exosomes are cell-derived small (30-90 nm) extracellular vesicles that promote intercellular communication and immunoregulatory functions. These “bioactive vesicles” shuttle various molecules, including miRNAs, to recipient cells. Inappropriate release of miRNAs from exosomes may cause significant alterations in biological pathways that affect disease development, supporting the concept that miRNA-containing exosomes could serve as targeted therapies for particular diseases. One of the most recent exciting findings is that microRNAs (miRNAs) exist in exosomes and these exosomal miRNAs can be functionally delivered to target cells. Thus, exosomal miRNAs are considered as biomarkers for many pathological states [refs]. Exosomes from diseased individuals contain miRNAs not found in normal, healthy subjects [refs].

We have inserted the following part of the above content into the introduction part: “Exosomes are cell-derived small (30-90 nm) extracellular vesicles that promote intercellular communication and immunoregulatory functions. These vesicles shuttle various molecules, including miRNAs, to recipient cells. One of the most recent exciting findings is that microRNAs (miRNAs) exist in exosomes and these exosomal miRNAs can be functionally delivered to target cells. Thus, exosomal miRNAs are considered as biomarkers for many pathological states [27, 28]. Exosomes from diseased individuals contain miRNAs not found in normal, healthy subjects [29, 30].”

27. Ciesla, M.; Skrzypek, K.; Kozakowska, M.; Loboda, A.; Jozkowicz, A.; Dulak, J. MicroRNAs as biomarkers of disease onset,  Bioanal. Chem.2011, 401, 2051-2061.

28. Michael, A.; Bajracharya, S.D.; Yuen, P.S.; Zhou, H.; Star, R.A.; Illei, G.G.; Alevizos, I. Exosomes from human saliva as a source of microRNA biomarkers, Dis.2010, 16, 34-38.

29. Taylor, D.D.; Gercel-Taylor, C. MicroRNA signatures of tumor-derived exosomes as diagnostic biomarkers of ovarian cancer,  Oncol.2008, 110, 13-21.

30. Rabinowits, G.; Gercel-Taylor, C.; Day, J.M.; Taylor, D.D.; Kloecker, G.H. Exosomal microRNA: a diagnostic marker for lung cancer,  Lung Cancer2009, 10, 42-46.

  1. What’s the “4” in line 65 before “Several previous studies…”? (오타) The other small suggestion is that there is no need to introduce too much experiment detail in the introduction, for example in line 80 “ …serum was introduced into the sensor array and incubated for 10 minutes…”.

Answer: Thank you very much for your point out and suggestion. I’m sorry, “4” is a typo. In addition, according to your suggestion, we have deleted the unnecessary part in line 80 of the introduction part as follows: “A mixed solution of three miRNAs in human serum was introduced into the sensor array and incubated for 10 minutes so that Each of the three miRNAs could hybridize with complementary capture nucleotides.”

  1. It will be clearer if list the sequences information in a table.

Answer: Thank you very much for your suggestion. We have tabulated the oligonucleotide sequences as follows:

Table 1. Oligonucleotide sequence used in this experiment.

Oligonucleotides

Sequences

miR-21

miR-106b

miR-155

capture probes (complementary sequence of miR-21)

capture probes (complementary sequence of miR-106b)

capture probes (complementary sequence of miR-155)

reference probe

universal detecting probe

reference detecting probe

5’-UAG CUU AUC AGA CUG AUG UUG A-3’

5’-UAA AGU GCU GAC AGU GCA GAU-3’

5’-UUA AUG CUA AUC GUG AUA GGG GUU-3’

5’-H2N-(CH2)6-TCA ACA TCA GTC TGA TAA GCT ACC CGG GCC CG-3’

5’-H2N-(CH2)6-ATC TGC ACT GTC AGC ACT TTA CCC GGG CCC G-3’

5’-H2N-(CH2)6-AAC CCC TAT CAC GAT TAG CAT TAA CCC GGG CCC G-3’

5’-H2N-(CH2)6-TTT TTT TTT T

5’-CGG GCC CGG G-(CH2)6-NH2-3’

5’-AAA AAA AAA A-(CH2)6-NH2-3’

  1. Supplement the details of probes immobilization and hybridization (Section 2.4 and 2.5). Better to draw the chemical reaction step by step, and how probes hybridize with each other with probes sequences.

Answer: Thank you very much for your suggestion. We have supplemented the more details of probes immobilization and hybridization (Section 2.4 and 2.5) as follows:

2.4. Immobilization of capture probes on SiO2 - coated SAW sensor array and conjugation of detecting probes on TiO2 nanoparticles

Capture probe immobilization on the sensor surface: Silicon dioxide (SiO2)-coated SAW sensor arrays were sequentially cleaned with deionized water and ethanol. They were then dried under a nitrogen atmosphere and activated in a UV-ozone chamber (144AX-220; Jelight Company Inc., Irvine, CA, USA) for 5 min, followed by incubation in 3 % (vol./vol.) 3-GPTES in ethanol for 1 h. After washing with ethanol and drying under nitrogen, they were baked at 110 °C in an oven for 1 h, washed again with ethanol, and dried under nitrogen. Next, 5'-amine-modified DNA capture probes were attached to the surface of 3-GPTES-modified SAW sensor arrays according to the following protocol: 3-GPTES-modified SAW sensor arrays were treated with 25 µM (concentration adjusted for high-density immobilization, Figure SX) of 5'-amine-modified three oligonucleotide capture probes and oligo T reference probe dissolved in 100 mM sodium phosphate buffer (pH 8.5) in separate spots for 60 min at 37 °C. The immobilization process of DNA capture probes on the SAW sensor chip was performed inside a humidity chamber to avoid evaporation. After washing with sodium phosphate buffer solution, the unreacted epoxide groups of 3-GPTES on the sensing area were deactivated by treatment with 50 mM 6-amino-1-hexanol in 100 mM sodium phosphate buffer (pH 8.5) for 30 min at 37 °C. Freshly modified sensor arrays were washed with sodium phosphate buffer and finally with doubly distilled water. Capture probe-modified SAW sensor arrays were desiccated at room temperature for storage until use.

Detecting probes conjugation on the TiO2 nanoparticles: The 3-GPTES modification process and conjugation process with 3'-amine-modified universal detecting probes (25 µM) and 3'-amine-modified oligo A reference detecting probe (25 µM) were the same as above. Detecting probe-modified TiO2 nanoparticles were stored in nuclease-free water at 4 °C until use.

2.5. Sandwich hybridization with photocatalytic silver staining

To detect miRNAs (miR-21, miR-106b and miR-155) using the SAW sensor array, 200 μL of a mixture of the equal concentrations of three synthetic miRNAs spiked in human serum was introduced to each sensor surface of SAW sensor array and allow to bind to the immobilized capture probes for 5 min at room temperature. In this step, a hybrid duplex between miRNAs and the corresponding capture probes were formed on the surface. This work was performed at various concentrations (0.1 pM to 1.0 mM) of miRNAs. After washing with sodium phosphate buffer solution for 1 min, a mixed solution of universal oligonucleotide detecting probe (1.0 mM) and oligo A’s reference detecting probe (1.0 mM) both conjugated with TiO2 nanoparticles in human serum was introduced to partially hybridized sensor surfaces and allowed to stand at 25 °C for 5 min for a complete sandwich hybridization. After washing with sodium phosphate buffer solution for 1 min, a silver nitrate (10 mM) in sodium phosphate buffer solution was added to each sensor surface and irradiated with UV light using a UV hand lamp with a wavelength of 365 nm (Vilber Lourmat, France) equipped with 4 W UV discharge tubes to induce silver staining reaction by photocatalytic reduction. After 2 min exposure, the SAW sensor array was then finally washed with sodium phosphate buffer solution. All experiments were repeated in quadruplicates.

Additionally, according to your suggestion, we used the probe sequences for each reaction step to draw how the probes hybridize to each other, as shown in Figure 4. (Figure 4 attached)

  1. For Figure 4, change the probes and target RNA with different color.

Answer: Thank you very much for your suggestion. We have modified the picture according to your suggestions. Please refer to Figure 4.

  1. The sandwich hybridization should be validated at least by gel assay. The length and concentration of probes should be optimized.

Answer: Thank you very much for your suggestion. The gel electrophoresis image of the sandwich hybridization assay (conducted for miR-106b) is shown in Figure S2, and we have included the optimization results of capture probe concentration in the Supplementary Information (Figure S1). (Figures attached)

  1. For Figure 5, there is no A label in the left top of the figure, the result of the sensor without target RNA should be tested.

Answer: Thank you very much for your point out and comments. We have inserted label “A” and a blank test without target miRNA has already been performed, and the resonance frequency change values ​​for all concentrations tested were blank subtracted. Thus, we have inserted the relevant information into the Section 3.2 and the captions of figure 5 as follows: “Here, blank refers to the result of all other processes proceeding without adding the target miRNA to the capture probe and the delta (Δ) frequency value of blank was 1.7 ± 0.15 KHz.” and “blank subtraction was performed on all delta (Δ) frequency values ​​for each concentration of miRNAs.”

  1. For table 1, this work should also be compared with other SAW based miRNA detection methods.

Answer: Thank you very much for your comments. Unfortunately, however, we did not find any previous SAW sensor-based miRNA detection studies. Thus, we have inserted the following into the Introduction part: “Researches have been conducted to analyze target DNA using SAW sensors [43, 44], but no studies on miRNA detection have been reported yet. Therefore, research on detecting miRNA using SAW sensor is of great significance and become the starting point of this study.”

  1. Huang, Y.; Das, P.Kr.; Bhethanabotla, V.R. Surface acoustic waves in biosensing applications, Actuators Rep. 2021, 3, 100041.
  2. Najla, F.; Mathieu, L.; Christophe, V.; Jean-Marie, F.; Chouki, Z.; Lionel, R.; Patrick, L.; Jean Jacques B.; and Christine, P. Surface Acoustic Waves Sensor for DNA-Biosensor Development, Lett. 2009, 7, 847-850.

In summary, I recommend this paper to be majorly revised before accepting.

Round 2

Reviewer 4 Report

Comments and Suggestions for Authors

Please answer all the questions and list the answers in a separate file with the revised manuscript.

Author Response

Please refer to the attached file, Response to Reviewers_Reviewer 4

Round 3

Reviewer 4 Report

Comments and Suggestions for Authors

I recommend this paper be accepted in its present form.